# A Review on Phase Field Modeling for Formation of η-Cu₆Sn₅ Intermetallic

Jia Sun [1], Lingyan Zhao [1], Huaxin Liang [1], Yao Zhang [2], Xuexiong Li [3], Chunyu Teng [4], Hao Wang [2] and Hailong Bai [1,*]

1 R&D Center of Yunnan Tin Group (Holding) Co., Ltd., Kunming 650106, China
2 Interdisciplinary Centre for Additive Manufacturing (ICAM), School of Materials and Chemistry, University of Shanghai for Science and Technology, Shanghai 200093, China
3 Institute of Metal Research, Chinese Academic of Science, Shenyang 100016, China
4 AVIC China Aero-Poly Technology Establishment, Beijing 100028, China
* Correspondence: baihl_ytc@163.com; Tel.: +86-13888160810

**Abstract:** Formation of intermetallic compounds (IMCs) exhibits remarkable microstructural features and provides opportunities for microstructure control of microelectronic interconnects. Excessive formation of brittle IMCs at the Cu/Sn interface such as η-Cu₆Sn₅ can deteriorate the reliability and in turn lead to solder joint failure in the Pb-free Sn-based solder joints. Phase field method is a versatile tool for prediction of the mesoscopic structure evolution in solders, which does not require tracking interfaces. The relationships between the microstructures, reliability and wettability were widely investigated, and several formation and growth mechanisms were also proposed for η-Cu₆Sn₅. In this paper, the current research works are reviewed and the prospective of the application of phase field method in the formation of η-Cu₆Sn₅ are discussed. Combined phase field simulations hold great promise in modeling the formation kinetics of IMCs with complex microstructural and chemical interactions.

**Keywords:** micro-interconnect; phase field model; intermetallic; η-Cu₆Sn₅





## 1. Introduction

With the rapid miniaturization of electronic circuits, thermal, electrical, and mechanical loads on three-dimensional integrated circuits (3DIC) that provide the electrical connection between the substrate and the chip have increased [1–4]. During the soldering process, a layer of IMC is formed at the solder–substrate interface. On the one hand, a thin, continuous, and uniform IMC layer is an essential requirement for good bonding; on the other hand, since this IMC layer is quite brittle, it can adversely impact the mechanical properties of the joint [5]. In the subsequent aging process, the microstructure of the solder joint coarsens while the interface IMC layer thickens, resulting in stress concentration, promoting the nucleation and growth of cracks. Additionally, as the IMCs generally have lowered ductility and higher electrical resistivity than pure Sn or Cu, the larger proportion of interfacial IMCs at the solder–substrate interface is a major reliability concern [6]. Since one of the essential features of solder technology is the reaction of a molten solder with a conductive metal to form an IMC, striking characteristics of IMCs have attracted considerable interests. Recently, the phase field method has been used as a powerful computational tool for modeling and predicting morphological and microstructural evolution in solder materials [7–10].

Based on the formation stages of the IMCs, there are mainly two types of IMCs which form during the interfacial reactions on the Cu/Sn/Cu sandwich structure that have been discovered and investigated, η-Cu₆Sn₅ and ε-Cu₃Sn. During the soldering process of Cu-Sn solder, the main reactions may happen as follows:

$$6Cu + 5Sn \rightarrow Cu_6Sn_5 \tag{1}$$

$$Cu_6Sn_5 + 9Cu \rightarrow 5Cu_3Sn. \tag{2}$$

Figure 1 shows the schematic diagram of the solder wetting process and scanning electron microscope (SEM) morphology of $Cu_3Sn$ and $Cu_6Sn_5$ IMCs. Generally, the soldering process can be divided into three stages, as Figure 1a shows [11]: (a) spreading; (b) base metal dissolution; (c) formation of an IMC layer during which $\eta$-$Cu_6Sn_5$ and $\varepsilon$-$Cu_3Sn$ nucleate and grow at the solder/conductor interfaces successively. Both $\eta$-$Cu_6Sn_5$ and $\varepsilon$-$Cu_3Sn$ can form during the interface reactions in the Cu-Sn binary system. Reactions such as $5Cu_3Sn \rightarrow 9Cu + Cu_6Sn_5$ and $2Cu_3Sn + 3Sn \rightarrow Cu_6Sn_5$ may also happen under different aging time and soldering conditions. The formation sequence of $\eta$-$Cu_6Sn_5$ and $\varepsilon$-$Cu_3Sn$ is still controversial at present since the experimental observations alone cannot give sufficient information to fully understand the growth kinetics and behavior of IMCs in solder joints [12–16].

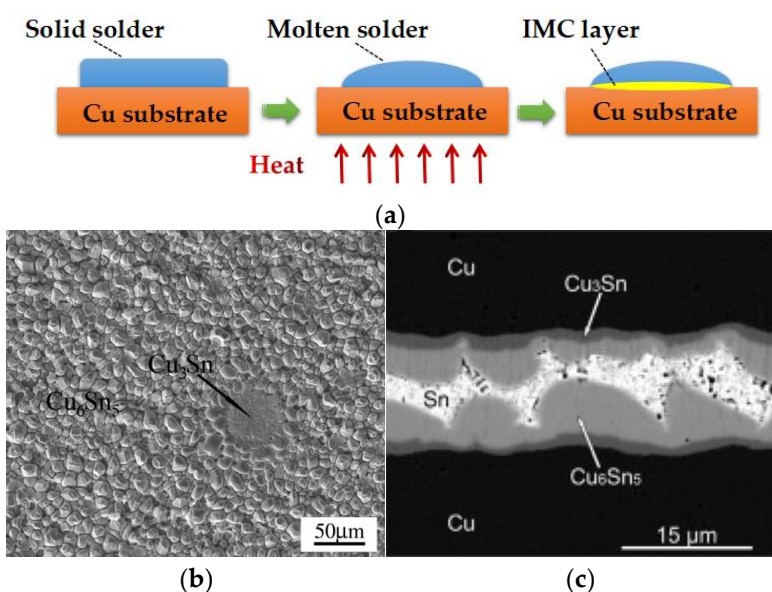

**Figure 1.** (**a**) Schematic diagram of solder wetting process; (**b**) top view of $Cu_6Sn_5$ and $Cu_3Sn$ grains formed on (001)Cu single crystal at reflow at 260 °C for 600 s [13]; (**c**) cross section view of Cu/Sn/Cu structure and IMCs [11].

$\eta$-$Cu_6Sn_5$ possesses a hexagonal structure (space group: P63/mmc) with $a_n = 4.190$ and it maintains orientation relationships with the Cu pad according to the edge-to-edge matching model. The two best matching close-packed planes that contain the rows of atoms are $(1\bar{2}10)_\eta \big|\big| (1\bar{1}1)_{Cu}$ and $(0\bar{1}10)_\eta \big|\big| (100)_{Cu}$, as illustrated in Figure 2 [16]. $Cu_6Sn_5$ is scallop-type and forms very fast during soldering process. The high-temperature $\eta$-$Cu_6Sn_5$ is first formed during the soldering process, and when the temperature is below 186 °C, $Cu_6Sn_5$ phase will transform $\eta$ into $\eta'$ ($\eta'$-$Cu_6Sn_5$), and the crystal structure will transform from hexagonal crystal to monoclinic crystal. The main object in this work is $\eta$-$Cu_6Sn_5$ that owns the hexagonal structure [5,14,15].

$Cu_3Sn$ forms between $Cu_6Sn_5$ and Cu if the contact of Cu substrates with the molten solder is long enough. In other words, $Cu_3Sn$ often nucleates and grows during the solder reflow process, as Figure 3 shows, and is also formed by diffusion and by reaction type growth as $Cu_6Sn_5$ [17]. Single crystalline Cu results in the growth of columnar $Cu_3Sn$ grains aligned in a thin uniform layer perpendicular to the interface, while a thick $Cu_3Sn$ layer formed from fine equiaxed grains on the polycrystalline substrate [18]. $Cu_3Sn$ or its interface with Cu ($Cu_3Sn$/Cu) is prone to voiding, either by the Kirkendall effect or by solute segregation, and thus more generally investigated in regard to the formation and growth of micro-voids [10].

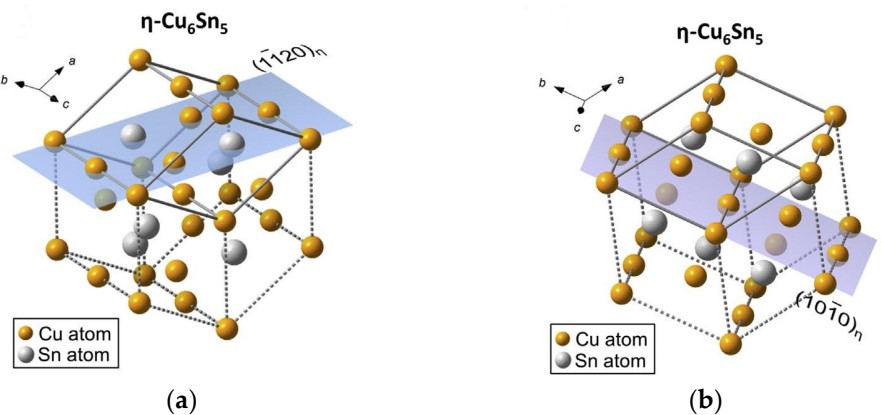

**Figure 2.** Schematic illustrations showing the 2D atomic site correspondence at the interface between (**a**) η-$Cu_6Sn_5$ and (001) Cu; (**b**) η-$Cu_6Sn_5$ and (111) Cu. The shaded area in the hexagonal lattice η-$Cu_6Sn_5$ shows the lattice plane in parallel with the Cu surface [16].

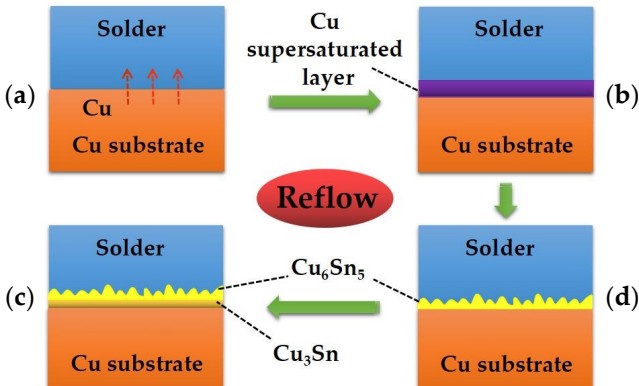

**Figure 3.** Scheme of the interfacial reaction of solder/Cu during solder reflow: (**a**) dissolution of the Cu substrate; (**b**) supersaturation of the molten solder layer with Cu; (**c**) formation of the scallop-type $Cu_6Sn_5$ at the interface; and (**d**) $Cu_3Sn$ emerges between $Cu_6Sn_5$/Cu with prolonged soldering.

Compared to $Cu_3Sn$, $Cu_6Sn_5$ is commonly the main research object due to its preferred formation, evident thickness, and thus greater effects on the microstructure and properties of solder joints [19,20]. Phase field simulations of IMCs' growth in lead-free solder joints have been reported in the literature, including diffusion and mechanical deformation. The formation kinetics of IMCs are quite complex and there are still some aspects which need to be clarified. Furthermore, detailed research work needs to be carried out for practical applications. In this paper, the recent developments in the application of the phase-field method in IMCs are reviewed and future directions for the research are proposed.

## 2. Phase Field Method for Formation and Growth of IMCs

### 2.1. Free Energy Curves of Cu(S), Sn(L) and IMCs

Generally, the multi-phase field model studying the evolution of microstructure involves three phases, Cu(S), liquid Sn(L), and η-$Cu_6Sn_5$ with multiple variants, whose structures are characterized by a set of non-conserved order parameters since the reaction $6Cu(S) + 5Sn(L) \rightarrow Cu_6Sn_5$ is diffusion controlled. The premise of most phase field modeling is to determine whether the phase transition process can occur, that is, whether the free energies of the reacting phases and their products have intersection in the relative phase diagrams [9,16,21–24]. Figure 4 shows the chemical free energy of Cu(S), liquid Sn(L), and η-$Cu_6Sn_5$ during reaction $6Cu(S) + 5Sn(L) \rightarrow Cu_6Sn_5$ under different conditions. The calculation of the free energy curves is based on the thermodynamic databases of the corresponding phases and IMCs from Thermo-Calc software, a powerful tool for the calculation

of phase diagrams according to different needs and purposes [25]. The energy curve of η-Cu₆Sn₅ with large curvature at $x_{Cu} = 0.545$ clearly reflects its stoichiometric feature.

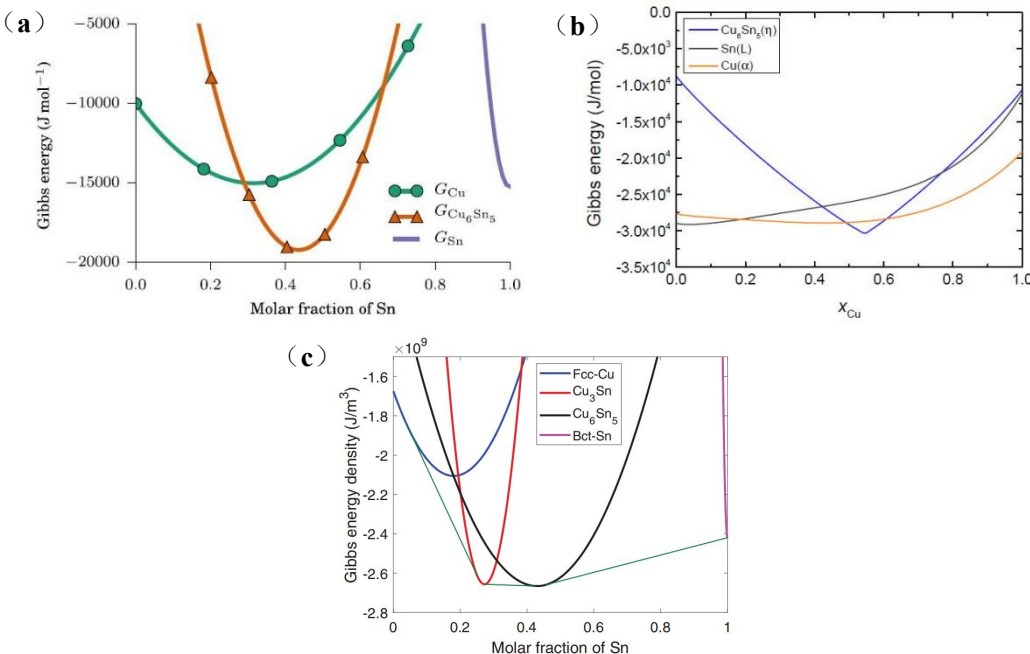

**Figure 4.** Plots showing the chemical free energy of Cu (s), liquid Sn (L), and η-Cu₆Sn₅ during reaction 6Cu + 5Sn→Cu₆Sn₅: (**a**) free energy evolution with Sn mole fraction as a function of composition at 25 °C [21]; (**b**) free energy evolution with Cu mole fraction as a function of composition at 250 °C [16]; (**c**) free energy evolution with Sn mole fraction as a function of composition in which condition both η-Cu₆Sn₅ and ε-Cu₃Sn precipitated [22].

### 2.2. Thermodynamic Parameters

The thermodynamic parameter formulas of the Cu-Sn binary system are summarized as follows according to the above free energy calculation database. It should be noted that the free energies of pure elements vary at different temperature ranges as noted in the parentheses following the relative expression. Table 1 shows the generally applicable thermodynamic parameters in most phase-field models for the formation and growth of IMCs in the Cu-Sn binary system [26]. What reactions happen and what phases or IMCs form depend on the specific conditions and needs.

**Table 1.** Gibbs free energy of phases in the Cu-Sn binary system (J·mol⁻¹) [26,27].

| Phases | Temperature Range | Free Energy |
|---|---|---|
| Sn (Pure) | 250 K < T < 505.08 K | $GHSERSN = -5855.135 + 65.443315 \times T - 15.961 \times T \times ln(T) - 0.0188702 \times T^2 + 3.121167 \times 10^{-6} \times T^3 - 61960 \times T^{-1}$ |
| | 505.08 K < T < 800 K | $GHSERSN = 2524.724 + 4.005269 \times T - 8.2590486 \times T \times ln(T) - 0.016814429 \times T^2 + 2.62313110^{-6} \times T^3 - 1081244 \times T^{-1} - 1.2307 \times 10^{25} \times T^{-9}$ |
| | 800 K < T < 3000 K | $GHSERSN = -8256.959 + 138.99688 \times T - 28.4512 \times T \times ln(T) - 1.2307 \times 10^{25} \times T^{-9}$ |
| Cu (Pure) | 298.15 K < T < 1357.77 K | $GHSERCU = -8256.959 + 138.99688 \times T - 28.4512 \times T \times ln(T) - 1.2307 \times 10^{25} \times T^{-9}$ |
| | 1357.77 K < T < 3200 K | $GHSERCU = -13542.026 + 183.803828 \times T - 31.38 \times T \times ln(T) - 3.642 \times 10^{29} \times T^{-9}$ |
| Sn (Liquid) | 100 K < T < 505.08 K | $GSNLIQ = 7103.092 - 14.087767 \times T + 1.47031 \times 10^{-18} \times T^7 + GHSERSN$ |
| | 505.08 K < T < 3000 K | $GSNLIQ = 6971.587 - 13.814382 \times T + 1.2307 \times 10^{25} \times T^{-9} + GHSERSN$ |
| Sn * (F.C.C.) | 298.15 K < T < 3000 K | $GSNFCC = 5510 - 8.46 \times T + GHSERSN$ |

**Table 1.** *Cont.*

| Phases | Temperature Range | Free Energy |
|---|---|---|
| Sn (B.C.C.) | 298.15 K $< T <$ 3000 K | $GSNBCC = 4400 - 6 \times T + GHSERSN$ |
| Sn (H.C.P.) | 298.15 K $< T <$ 3000 K | $GSNHCP = 3900 - 7.646 \times T + GHSERSN$ |
| Cu (Liquid) | 298.15 K $< T <$ 1357.77 K <br> 1357.77 K $< T <$ 3200 K | $GCULIQ = 12964.736 - 9.511904 \times T - 5.849 \times 10^{-21} \times T^7 + GHSERCU$ <br> $GCULIQ = 13495.481 - 9.922344 \times T - 3.642 \times 10^{29} \times T^{-9} + GHSERCU$ |
| Cu (B.C.C.) | 298.15 K $< T <$ 3000 K | $GCUBCC = 4017 - 1.255 \times T + GHSERCU$ |
| Cu (B.C.T.) | 298.15 K $< T <$ 3000 K | $GCUBCT = 4184 + GHSERCU$ |
| $Cu_6Sn_5$ | – | $G = -7085.92 + 0.15558 \times T + 0.545 \times GHSERCU + 0.455 \times GHSERSN$ |
| $Cu_3Sn$ | – | $G = -8194.2 - 0.2043 \times T + 0.75 \times GHSERCU + 0.25 \times GHSERSN$ |

\* Many reactions such as L $\rightarrow$ Sn + $Cu_6Sn_5$ involve the metastable phases in the Cu-Sn system under the interactions of thermal, electrical, and force fields. Therefore, FCC and BCT in the above table respectively represent face-centered cubic or body-centered foursquare phases.

Thermodynamic parameters in the Cu-Sn binary system are present in Table 2. It can be seen that these parameters (they correspond to the "$L_{ij}$" in the phase field model as mentioned in Section 2.3.2) are only relative to temperature. These results may have slight errors due to artificial calculation or measurements which is a normal phenomenon and does not affect the application of the data.

**Table 2.** Thermodynamic parameters in Cu-Sn binary system [27].

| Phases | Thermodynamic Parameters |
|---|---|
| Liquid | $L0_{Cu:Sn}^{liq} = -9002.8 - 5.8381 \times T$ <br> $L1_{Cu:Sn}^{liq} = -20,100.4 + 3.6366 \times T$ <br> $L2_{Cu:Sn}^{liq} = -10528$ |
| F.C.C | $L0_{Cu:Sn}^{fcc} = -11,106.95 + 2.0791 \times T$ <br> $L1_{Cu:Sn}^{fcc} = -15,718.02 + 5.92547 \times T$ |
| B.C. T | $L0_{Cu:Sn}^{bct} = 21,000$ |

Other solder specific properties such as diffusivity, mobility, and interface energy must be introduced into the models through phenomenological parameters, which are determined based on experimental and theoretical information.

### 2.3. Phase Field Modeling

#### 2.3.1. Nucleation of IMCs

A proper nucleation mechanism is necessary for the nucleation of IMCs due to the field theory of self-consistent phase fields. There are mainly two types of nucleation in the formation process of IMCs in micro-interconnects: classical nucleation [20] and noise nucleation [16]. Classical nucleation theory applies discrete Poisson probability distributions to describe the probability of nucleation events at a particular point in time or space. This method was originally proposed by Simmons [28] and used in the classical phase field equation to replace the Langevin noise. The probability for the (heterogeneous) nucleation of a new phase at an arbitrary point in space and time is determined by thermodynamic and kinetic factors such as local driving forces for the nucleation of the new phase as well as interfacial energies that act as barriers to the nucleation. The stochastic nature of nucleation is then approximated by unity "minus" the zero-event probability of a Poisson distribution,

which is applied to replace the Langevin noise [29]. The expression of Poisson nucleation probability $P^n$ and nucleation rate $I$ are as follows [22,23]:

$$P^n = 1 - exp[-I.v.\Delta t] \tag{3}$$

where $\Delta t$ is the time interval over which the probability of nucleation is to be determined, while $v$ represents the volume over which the nucleation probability is calculated [30]:

$$I = I_0 exp\left[-\frac{16\pi\sigma^3}{3k_B T(\Delta G_V)^2}\frac{cos^3\theta - 3cos\theta + 2}{4}\right] \tag{4}$$

where $\sigma$ is the interface energy, $\Delta G_V$ is the nucleation barrier, $\theta$ is the contact angle, and $I_0$ is the nucleation rate constant. Equation (4) is the classical theoretical formula of IMCs' nucleation in micro-interconnect. The nucleation of IMCs is determined by probability. Noise nucleation refers to Langevin or stochastic noise terms to describe the thermal fluctuation of order parameters, which is usually appended to the end of the phase field evolution equation. In general, the noise term $\xi$ follows a Gaussian distribution and satisfies the fluctuation dissipation theorem [31]:

$$\langle\xi(r_1,t_1)(r_2,t_2)\rangle = -2k_B T\nabla\cdot\Gamma_{ij}\nabla[\delta(r_1 - r_2)\delta(t_1 - t_2)] \tag{5}$$

$$\langle\xi(r_1,t_1)(r_2,t_2)\rangle = 2k_B TL_{ij}\nabla[\delta(r_1 - r_2)\delta(t_1 - t_2)] \tag{6}$$

where $k_B$ is the Boltzmann constant, $T$ is the temperature, and $\Gamma_{ij}$ is the symmetric part of the mobility $M_{ij}$. The above equations correspond to non-conservative and conservative field variables, respectively. The origin of the noise term is related to the microscopic degrees of freedom, such as thermal vibrations, which transforms the statistical expression back into microscopic form. The nucleation of IMCs will be random as one chooses the noise nucleation method.

### 2.3.2. Evolution Equations

A phase field model must first describe the microstructure by using a set of conserved and/or non-conserved field variables. Second, the evolution of these field variables must be determined by applying the temporal and spatial evolution of the field variables. The morphology evolution and growth kinetics of IMCs with interfacial reaction of the Cu-Sn binary system were studied by the multi-phase field model. Phase field models on IMCs in solders are generally derived from the work of Steinbach [32–34] et al., which are applicable to the formation and growth of IMCs in micro-interconnects. In the $6Cu + 5Sn \rightarrow Cu_6Sn_5$ interface reaction, a general model of the total free energy of a chemically heterogenous system that involves interfacial, bulk, elastic, and electrical interaction is [16,34]

$$F^{tot} = \int f^{bulk} + f^{int} + f^{elas} + f^{elec} \tag{7}$$

where $f^{bulk}$, $f^{int}$, $f^{elas}$, and $f^{elec}$ denote the chemical free energy, interfacial energy, elastic energy and electric field energy of the system, respectively. One can also expand thermal gradient field $f^{therm}$ [7], magnetic field $f^{mag}$ [35], and plastic field $f^{plas}$ [21] according to the demand, which are not the main objects in this work. $f^{bulk}$ and $f^{int}$ are temperature-dependent and represent the temperature field. Usually, these two terms are expressed in the same phase field evolution equation. Moreover, the Kim–Kim–Suzuki (KKS) model is employed in the phase field model by considering the interface as a mixture of phases with equal chemical potentials [16] ($f^{bulk} + f^{int}$), since the formation of IMCs is a second phase precipitation process controlled by diffusion. Additionally, the elastic field $f^{elas}$ is usually represented in the simulation of the formation kinetics of $Cu_6Sn_5$ to solve the problem of different variation evolution laws, since the phase field simulation cannot solve the plastic deformation problem at present. $f^{elec}$ is the electrical field at last.

The chemical free energy density of Cu and liquid Sn are available in the binary CALPHAD database [36] and can be expressed as

$$f_{ch}^{\rho} = \sum_{i=Cu,Sn} x_i^0 G_i^{\rho} + RT \sum_{i=Cu,Sn} x_i ln x_i + G_E^{\rho} \tag{8}$$

where $x_i$ is the conserved order parameter or atomic fraction of species $i$, and $iG_i^{\rho}$s the Gibbs energy of the pure element $i$ in the chemical phase $r$. The last term corresponds to the excess Gibbs energy, which can be written by the Redlich–Kister polynomial [27]:

$$G_E^{\rho} = x_{Cu} x_{Sn} \sum_{r=0}^{n} \left[ L_{Cu,Sn}^{\rho,r} (x_{Cu} - x_{Sn})^r \right] \tag{9}$$

where $L_{Cu,Sn}^{\rho,r}$ is the temperature-dependent interaction parameters in the binary system.

Due to the stoichiometric nature of η-Cu$_6$Sn$_5$, the chemical free energy was described as a temperature dependent single value instead of a continuous function of composition [37,38]. The description causes a difficulty for any phase field model because the variational derivative of free energy functional with respect to composition is not available. To resolve the issue, many investigators follow the work of Moelans [38–41] by treating IMCs such as η-Cu$_6$Sn$_5$ and ε-Cu$_3$Sn as a chemical compound with small solubility and adopting a two sublattice model $(Cu, Sn)m(Sn, Cu)n$. The chemical free energy of η-Cu$_6$Sn$_5$ is expressed based on the compound energy formulation [18]:

$$f_{ch}^{\eta} = \sum_{i,j=Cu,Sn} y_i^I y_j^{II} G_{i:j}^{\eta} + RT \left( m \sum_{i=Cu,Sn} y_i^I ln y_i^I + n \sum_{i=Cu,Sn} y_i^{II} ln y_i^{II} \right) \tag{10}$$

where $y_i^I$ and $y_i^{II}$ are site fractions of species $i$ in sublattice $I$ and $II$, respectively, $G_{i:j}^{\eta}$ is the Gibbs energy of formation of the compound $i_m j_n$ or the so-called end member phases, and $m$ and $n$ give the ratio of sites on the two sublattices, which are 0.545 and 0.455, respectively.

The order parameters are constrained by $\sum_i \varphi_i = 1$. The total free energy functional is represented as [39]

$$f^{bulk} + f^{int} = \int \left\{ \sum_{i,j(i<j)} \left[ \frac{\kappa_{ij} a_{ij}^2}{2} (\varphi_j \Delta \varphi_i - \varphi_i \Delta \varphi_j)^2 + \omega_{ij} \varphi_i \varphi_j \right] + \sum_i \varphi_i \frac{f_{ch}^i (x_{ch}^i)}{V_m} \right\} dV . \tag{11}$$

The governing equation of non-conserved order parameters was derived as [7]

$$\frac{\partial \varphi_i}{\partial t} = -\frac{2}{N_p} \sum_{j(j \neq i)} s_i s_j a_{ij}^2 L_{ij} \left( \frac{\delta F_{tot}}{\delta \varphi_i} - \frac{\delta F_{tot}}{\delta \varphi_j} \right) + \xi(\vec{r}, t) \tag{12}$$

where $N_p$ is the local number of coexisting phases, $s_i$ is a step function which is equal to 1 when the phase $i$ exists and is otherwise equal to 0, $L_{ij}$ is the mobility of the interface between phases $i$ and $j$, and z is the Langevin noise terms describing thermal fluctuations of the order parameters [10]. The time-evolution of the conserved order parameter or atomic fraction is governed by the Cahn–Hilliard equation:

$$\frac{\partial C}{\partial t} = V_m^2 \nabla \cdot \left[ \widetilde{M} \nabla \left( \frac{\delta F_{tot}}{\delta c} \right) \right]. \tag{13}$$

The elastic strain energy functional is formulated by the micro-elasticity theory derived from Hooke's law [16]:

$$f^{elas} = \frac{1}{2} \int C_{ijkl} \varepsilon_{ij}^* \varepsilon_{kl}^* dV + \frac{V}{2} C_{ijkl} \bar{\varepsilon}_{ij} \bar{\varepsilon}_{kl} \bar{\varepsilon}_{ij} \int C_{ijkl} \varepsilon_{kl}^* dV - \frac{1}{2} \int \frac{d^3 g}{(2\pi)^3} n_i \widetilde{\sigma}_{ij}^* \Omega_{jk}^0 \widetilde{\sigma}_{kl}^* n_l \tag{14}$$

where $C_{ijkl}$ is the elastic stiffness tensor, and $\varepsilon_{ij}^*$ is the eigenstrain.

The term in square brackets represents the negative atomic flux of Cu defined in the laboratory frame of reference and $\tilde{M}$ is the corresponding chemical mobility.

The electrical field is expressed as [42]

$$f^{elec} = N_A e\psi \sum_i Z_i^* c_i \tag{15}$$

where $N_A$ is the Avogadro number, $e$ is the natural charge of an electron, $\psi$ is the electronic potential, and $Z_i^*$ is the effective nuclear charge of the phases.

Equations (11), (14) and (15) correspond to terms in Equation (7) of total free energy, respectively, which is the phase field equation representing the evolution of IMC Cu$_6$Sn$_5$ in micro-interconnect. Parameters such as m and n, free energy, and interaction coefficient $L_{ij}$ can be modified as it is used to simulate the formation of other IMCs.

Figure 5 shows the schematic flow chart of the formation mechanism simulated by the phase field method. Firstly, the free energy curves of Cu, Sn, and the relative IMC should be obtained to ensure that the reaction that produces η-Cu$_6$Sn$_5$ and ε-Cu$_3$Sn can happen; secondly, a proper nucleation should be chosen according to the conditions and needs; then, appropriate thermodynamic parameters needs to be chosen to establish a proper simulation model of IMCs; finally, the thermal, stress, and electrical field can be coded one by one in the phase field model.

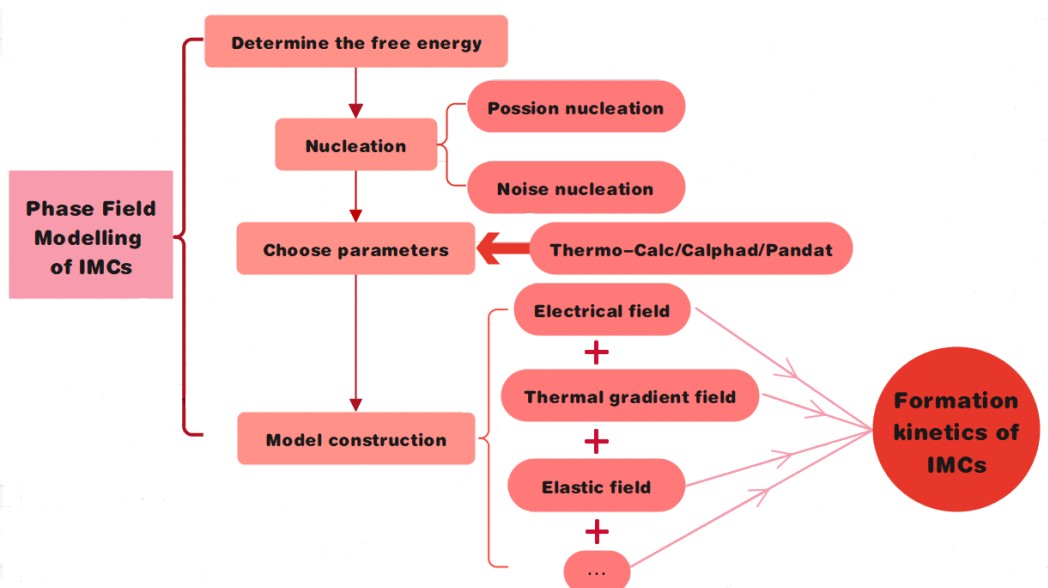

**Figure 5.** Flow chart of kinetics of formation of IMCs by the phase field method.

Park [23] et al. investigated the evolution of IMC layers in the Cu/Sn solder system under electromigration conditions by a multi-phase field formalism through considering the concurrent evolution of the Cu$_3$Sn and Cu$_6$Sn$_5$ IMC layers. It turned out that back- stress resulting from the non-equilibrium vacancy generation due to diffusive flux imbalance among the individual diffusants (Cu and Sn) played a fundamental role in the evolution of the IMC layers, and the simulations and experiments showed nice qualitative agreement. It can be deduced from their work that there are interactions between micro- voids and IMCs, and they interact with each other due to the diffusion of Cu and Sn under thermal–electrical–stress effects, which is really complex in 3DIC.

Ke [16] et al. investigated pattern formation during interfacial reactions in between Liquid Sn and two types of unidirectional Cu substrates, (001) and (111), by using a combination of crystallographic analysis, the theory of micro-elasticity, and phase field simulations. The simulation results showed that each symmetry-related η-Cu$_6$Sn$_5$ variant grew along the direction of best atomic site matching with the substrates. Simulation predic-

tions in their work also demonstrated excellent agreements with experimental observations. It can be concluded from their investigation that η-$Cu_6Sn_5$ variants will precipitate along the fit directions which take on the lowest energy for their stable characteristics. This team also found that heteroepitaxial constraints led to spatial alignment of η-$Cu_6Sn_5$ variants, which may shed light on the fact that most research works on η-$Cu_6Sn_5$ did not show variant features, under which condition no heteroepitaxial growth took place.

Kunwar [7] et al. integrated neural network analysis with the multi-phase field method to quantify the mechanism of thermo-migration at the cold side of a solder-substrate system. Data on heats of transport, temperature difference, and the growth rate constant of IMCs were obtained from multiple FEM (Finite Element Modeling) simulations, and the FEM-generated dataset was employed in the neural network. The growth rate constant predicted by machine learning was tallied with the experimental values. The heat conduction and phase field equations were solved using FEM which was innovative and more efficient than the phase field calculation. However, it should be noted that the accuracy of the dataset generated by FEM is still questionable before it is employed in the neural network.

Durga [22] et al. used a phase field model for multi-phase systems that can treat diffusion, elastic, and especially plastic deformation to study the problem of the growth of intermetallic phases in solder joints undergoing mechanical deformation. Reuss–Sachs and Voigt–Taylor models were combined to describe the plastic part, which was essentially superior to the Khachaturyan scheme applied by most previous studies. It is well known that the behaviors of real materials probably lie between full inheritance of plastic deformation and no inheritance at all. The plastic part in their work necessitated the development of models allowing for the partial inheritance of plastic deformation, which can be adapted in future studies that need to couple the plastic deformation in phase field model of IMCs' growth.

In addition, phase field method can also be used to study the wedding mechanism of Sn-Bi alloys [43], the void growth behaviors of lead-free solder alloys [44], phase separation in Sn-Ag-Cu solders [45], etc.

## 3. Future Research Aspects

### 3.1. Interactions between IMCs and Micro-Voids

The structure and morphology of the $Cu_6Sn_5$ and $Cu_3Sn$ IMC layers should be investigated quantitively since they control the integrity of microelectronic interconnects. At the same time, the formation and accumulation of vacancies in $Cu/Cu_3Sn$ interfaces also play a key role in the reliability of 3DIC devices. In fact, IMCs and micro voids generally form concurrently due to the combined functions of thermal, electrical, and stress fields. In other words, there must be interactions between IMCs and vacancies, which is the absolute factor affecting solder joint reliability. In the phase field model, the interactions between IMCs and voids can be coupled together in spite of their distinguished sizes, positions, and characteristics if they are under the same thermal, electrical, and stress fields. Clarity of the interactions between IMCs and voids can be helpful to further elucidate the mechanism of solder joint failure led by interfacial phases and vacancy evolution.

### 3.2. Couple Multiple Simulation Methods

Multi-scale modeling is a trend in the field of computational material science, not only in the application of the phase field simulation in formation of IMCs and vacancies. The first-principles calculations can provide plastic/elasticity tensors and potential energies; the diffusion number of Cu and Sn atoms during the soldering process can be obtained by molecular dynamics; one can obtain the stress and strain distribution by FEM calculation under the thermos–electro–mechanical fields through which the formation site of IMCs and voids can be predicted. The combination of these approaches will provide a better estimation for the lifetime of solder joints. These simulation methods all can be properly coupled with the phase field models to interpret and predict more deeply the formation

and growth of IMCs; they can provide information that cannot be obtained in the phase field simulation alone. Which method is chosen depends on the practical use.

### 3.3. Quantitative Prediction of Growth of IMCs

The future phase field model should be able to quantitatively capture the kinetic features such as the volume and thickness of the IMC layer according to conditions such as temperature, electron wind, or/and electric current density so as to control the proper thickness of IMCs in industrial production. It is also clear that plastic effects must be considered in the phase field model to obtain realistic results. Additionally, since, in phase field simulations, the width of the interfaces is usually artificially enlarged to limit the computational requirements, the effect of this excess energy can become excessively high which is inconsistent with reality. The difficulty in narrowing the gap between the simulated results and the actual situation should be overcome.

Moreover, since the thickness of the IMC layer can be reduced by adding stabilizing elements such as Al [46] and nanoparticles such as $CeO_2$ [47] in the tin solder, future phase field simulations or calculations should consider building a multi-elements system coupling other alloying elements and additives to interpret mechanisms. Relative phase field simulation can shed light on the growth kinetics of interactions between alloying elements and IMCs, which cannot be achieved by experimental tools alone. Otherwise, the quantitative relationship between alloying elements and IMCs should also be considered so as to obtain more actual results that can serve as a guide to experiment and production.

### 4. Conclusions

Phase field modeling on formation and growth of IMCs generally follows four steps: (1) determine the free energy curves of the reaction phases and the productive phases; (2) choose proper thermal dynamic parameters; (3) determine a fit nucleation mechanism; (4) construct the evolution equations on thermal, electrical, and stress fields, etc. The fourth step can also be divided into several sub-steps, approached one by one, in order to make sure that each step is reasonable and correct. The sequence of second and third steps can be exchanged since this will not affect the modeling. In order to achieve more accurate results in phase field simulations of IMCs, their interactions with micro voids and other simulation approaches should be considered and they are both promising aspects.

**Author Contributions:** Conceptualization, H.L. and J.S.; methodology, J.S.; validation, J.S. and X.L.; formal analysis, L.Z. and H.W.; investigation, L.Z. and J.S.; writing-original draft preparation, J.S. and Y.Z.; writing-review and editing, J.S. and C.T.; visualization, J.S.; supervision, H.B.; project administration, H.B. and L.Z.; funding acquisition, H.B. and L.Z. All authors have read and agreed to the published version of the manuscript.

**Funding:** This research was funded by Applied Basic Research Foundation of Yunnan Province, grant number 202101BC070001-010 and Chuncheng Project III.

**Data Availability Statement:** The data presented in this study are available upon request from the corresponding author. The data are not publicly available due to technical or time limitations.

**Acknowledgments:** The authors are grateful to Hao Wang, Lingyan Zhao, and Hailong Bai for their contribution to the discussion.

**Conflicts of Interest:** The authors declare no conflict of interest.

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
