# Peer review of "A Review on Phase Field Modeling for Formation of η-Cu6Sn5 Intermetallic"

_metals, doi:10.3390/met12122043_

Round 1

Reviewer 1 Report

The maniscript „A review on phase field modeling for formation of η-Cu6Sn5 in-2 ter-metallic compound“ provide an interesting summary of knowledge regarding phase field modeling on formation and growth of IMCs at the Copper/Sn solder boundaries. The paper provide a significant contribution to the field of interconections in soldering and electronics.

The paper is well organized, provides interesting findings, however, there should be considered to add

a paragraph with relevant references that also considers the possibilities of reducing the thickness of the IMC by adding stabilizing elements to the tin solder.

You can draw on this issue, for example, from this publication:

Comparison study of SAC405 and SAC405+0.1% Al lead free solders. In Soldering and Surface Mount Technology. Vol. 25, Iss. 3 (2013), s.175-183. ISSN 0954-0911

This issue is very actual nowadays, and the simulations or calculations should consider stabilizing elements into account.

Author Response

Dear Reviewer,

    Thanks a lot for your comments and advice!

    We added a paragraph to propose the prospective on the application of phase field method on the IMCs and additives such as stabilizing elements or nano-particles. “Comparison study of SAC405 and SAC405+0.1% Al lead free solders” is really a masterpiece. We have cited it in this paper and will use it in our future works.

Best regards.

Reviewer 2 Report

This paper reviews phase field modeling regarding the formation of Cu6Sn5 IMCs. Its explanation contains sufficient information and would be very helpful for related readers of the journal. The authors also present future expected research aspects from the point of view of microelectronic interconnections. It would be accepted for publication in the present form.

Author Response

Dear Reviewer,

    Thanks a lot for your comments and advice! 

    We will continue to work hard in the phase field modelling of IMCs in Tin solders.

Best regards.

Reviewer 3 Report

1. The list of keywords need to be expanded.

2. The whole article is about eta-Cu6Sn5 (with HCP crystal structure), which is a thermodynamically stable phase at T greater than 459 K. At T less that 459 K, eta'-Cu6Sn5 (monoclinic structure) is stable. If eta phase is mentioned, it should be distinguished from its low temperature  twin (eta' phase) in the introduction section.

3. An eta-Cu6Sn5 is an intermetallic compound, and linguistically represented not as intermetallic compounds.

4. Nucleation modeling must be transferred to the section outside phase field modeling.

5. The expression for the equation of total free energy must be generalized or expanded to include free energy corresponding to magnetic field, thermal gradient field etc.

6. The purpose of KKS model is to enable multiphase modeling in a thermodynamically consistent modeling. (E.g. equality of chemical potential between adjacent phases, relation between global and local concentration fields).

The authors are requested to elaborate further about the meaning of "KKS model is used to establish the thermal field model".

Author Response

Dear Reviewer,

    Thanks a lot for your time and attention! 

    Your  comments were all very helpful and we have followed your advice and corrected the improper parts. We will continue to work hard in the phase field modelling of IMCs in Tin solders and learn from masters like you!

Best regards.

Round 2

Reviewer 3 Report

Expansion of the list of keywords: It is recommended to include one or two additional keyword(s) so that the total number of keywords is 4 or 5.